# Is the Teaching Environment a Risk Factor for Depression Symptoms? The Case of Capricorn District in Limpopo, South Africa

Khomotso Comfort Maaga [1,*] and Kebogile Elizabeth Mokwena [2,*]

1   Department of Public Health, Sefako Makgatho Health Sciences University, Pretoria 0208, South Africa
2   NRF Chair in Substance Abuse and Population Mental Health, Sefako Makgatho Health Sciences University, Pretoria 0208, South Africa
*   Correspondence: khomotso.maaga@smu.ac.za (K.C.M.); kebogile.mokwena@smu.ac.za (K.E.M.)

**Abstract:** The global increase in mental disorders also identifies the workplace, including the teaching environment, as a key source of such disorders. Social problems among learners often put additional pressure on the teachers, over and above their normal academic, administrative and organizational responsibilities, thus contributing to high levels of stress among teachers. The purpose of this study was to determine the prevalence of depression symptoms, as well as the associated sociodemographic factors, among teachers in Capricorn District, Limpopo Province, South Africa. A cross-sectional quantitative study design using the Patient Health Questionnaire (PHQ-9) tool was used to determine the symptoms of depression among a sample of 381 teachers. A self-administrated questionnaire was used to collect sociodemographic data, which were analyzed descriptively. Pearson chi-square tests were used to explore associations between a range of sociodemographic variables and PHQ scores. A final logistic regression model was used for factors that were significantly associated with depression symptoms according to Chi-square tests. The majority of the participants were Black (83.45%) and female (70.87%) and had obtained a bachelor's degree as their highest qualification (53.95%). Almost half of participants (49.87%) tested positive for symptoms of depression, which ranged from mild to severe. Employment-related factors that were significantly associated with depression symptoms included the quintile ranking of the school, the school where employed, learner-to-teacher ratio and the subjects taught by the teacher. Personal factors that were associated with depression included gender, marital status and race. Depression symptoms amongst teachers were mostly associated with workplace factors.

**Keywords:** depression; educators; teachers; South Africa; Patient Health Questionnaire (OHQ-9)

## 1. Introduction

The global burden of a range of mental disorders, including depression, is reported to be increasing, with public health concerns, as depression is a debilitating disorder that compromises quality of life, increasing disability and, in some cases, even death [1]. However, there is dearth of studies conducted in low- and middle-income countries, including South Africa, on its burden and contributory factors. Therefore, there is a need for studies in this area of public health to quantify the rates and severity of depression in the population, which can provide information on prevention and treatment of such disorders, in addition to promoting mental health. Depression is a mental health disorder that negatively affects the mood of an individual and is accompanied by symptoms such as loss of enjoyment of life, irregular sleeping patterns, poor eating habits, prolonged periods of excessive sadness, and feelings of hopelessness and worthlessness. The World Health Organization estimates the global prevalence of depression to be 3.8% [1]. The workplace has been identified as a key contributory factor to mental disorders, including depression [2], due to its highly demanding and stressful nature.

Across the globe, the teaching environment has been associated with mental health distress problems such as burnout, depression and anxiety [3,4]. These mental health challenges among teachers are reported in both developed and developing countries, including in African countries such as Egypt and Tanzania [5,6], as well as countries such as Malaysia, China and the USA [7–9]. Depression has also been reported to be more prevalent among younger, less experienced and female teachers [5,6,10–12]. The working conditions within the education sector have left many teachers dissatisfied with their jobs, as evidenced by the high turnover rates among teachers [13].

The South African school system consists of primary (grades R-7) and high (grades 8–12) schools, which, in most cases, are separate. However, some schools consist of all grades (i.e., grades R-12), which are referred to as combined schools. The system is further divided into private and public schooling; private schools are independently run and funded, and public schools are funded by the government. South African public schools are further classified according to a quintile ranking system, with lower-quintile-ranking schools (Q1–Q3) being no-fee-paying schools, whereas upper-quintile schools (Q4–Q5) are classified as fee-paying schools. This ranking system is determined by the socioeconomic conditions within the neighborhoods in which the schools are situated [14]. Anecdotal evidence suggests that teacher employment responsibilities, as well as the social environment of the area in which they work, are potential risk factors for mental disorders, including depression.

Over and above teaching responsibilities, teachers are also responsible to other organizational tasks that are not directly related to teaching, which include preparing the learners for concerts, excursions, sporting events and other extracurricular activities that come with added responsibilities. In South Africa, teachers are faced with social ills within the country that spillover into the teaching environment and inevitably heighten their job demands. Specifically, orphaned and other vulnerable children often have social problems such as cases of child abuse [15], teenage pregnancy [16] and widespread issues associated with HIV/AIDS [17], which require extra attention and support from educators. This necessitates teachers taking on caregiving responsibilities as the distortion of roles often puts extra workload on educators to perform responsibilities that go above and beyond just teaching. This burden is mostly felt by public school teachers, which would explain why the literature has reported higher rates of depression among public school teachers than those in working the private sector [18]. The school level and subjects taught by teachers have also been connected to higher rates of depression, as higher rates of depression were reported among teachers of commercial and science subjects [18]. Although some authors have reported higher rates of depression among primary school teachers [5], high school teachers carry a disproportionate amount of stress compared to primary school teachers [19], which is largely attributed to the behavior of the learners, which progressively worsens during adolescent years.

Cases of misbehavior and ill-discipline among learners, which often include substance abuse [18] and increasing violence against teachers by students [19,20], challenge the mental well-being of teachers and often result in depression. Furthermore, the lack of resources in low-socioeconomic areas and schools results in work environments that are not conducive to optimum mental health status because teachers are required to improvise to achieve the intended educational outcomes. Limited resources and a lack of infrastructure make it difficult for teachers to carry out their duties efficiently, increasing their risk of mental distress [2,21]. Additionally, the consequences of the COVID-19 pandemic have further contributed to the compromised mental health of educators, as reports of increases in depression during and after the pandemic have emerged [22–24]. In particular, teachers are often required to make up for lost time due to any disruptions that occur in the teaching and learning processes, placing a lot of pressure on them.

Depression among teachers transcends the individual teacher and affects the health, well-being and development of the students under their care. Depression among teachers was positively associated with psychological distress amongst students [25], which translates to compromised academic [26], behavioral [27] and social adaptation [28] among

students. Additionally, teachers with mental health disorders such as depression, whether diagnosed or not, have higher rates of absenteeism and presentism, which relates to teachers being at work but not adequately productive because they are physically or mentally unwell [29]. Both absenteeism and presentism lower productivity in the teaching environment and compromise the quality of teaching and learning. This study was conducted on the basis of these premises in order to determine the prevalence of depression amongst teachers, as well as contributory factors, in the Limpopo Province, South Africa.

## 2. Purpose of the Study

The purpose of the study was to screen for depression symptoms and associated sociodemographic and employment-related factors among teachers in Capricorn District, Limpopo Province. It was expected that the prevalence of depression symptoms would be high and that differences in sociodemographics of the individual teachers (e.g., gender, age, race and highest level of education), as well as work environment factors (e.g., student-to-teacher ratio, employment status, subjects taught and working experience) across schools, would produce variations in the results of the study.

## 3. Methodology

### 3.1. Study Design

A cross-sectional study design was adopted, using self-administrated questionnaires.

### 3.2. Research Setting

The study was conducted in two local municipalities (i.e., Polokwane and Lepelle-Nkumpi, South Africa) within Capricorn District, Limpopo Province, South Africa. Capricorn District has 541 primary schools and 342 high schools under four local municipalities: Polokwane, Molemole, Lepelle-Nkumpi and Blouberg. The district stretches across both urban and rural areas.

### 3.3. Population, Materials and Procedure

The population consisted of primary and high school teachers in both public and private schools in Limpopo Province, South Africa.

Recruitment of participants started by obtaining permission to conduct the study from the Limpopo Department of Education in Polokwane, which was used to negotiate for permission from the municipalities of Polokwane and Lepelle-Nkumpi. The obtained permission letters were used to negotiate for permission from the management of the identified schools. Within the selected schools, a survey was conducted in which all teachers were recruited to participate in the study. Data were collected by the researcher over a period of seven months, from January 2022 to July 2022.

A researcher-developed questionnaire was used to collect sociodemographic data from the participants. The globally validated Patient Health Questionnaire (PHQ-9) was used to screen for symptoms of depression. The tool asks participants about depression-related symptoms over the previous two weeks, with options ranging from "0" (not at all) to "3" (nearly every day). It asks questions related to the participant's sleeping and eating patterns, ability to concentrate on tasks, thoughts related to self-harm, etc. The maximum score is 27, and a score of 5 and above is a positive indicator of depression symptoms, with higher scores indicating higher levels of depression symptoms. The psychometric properties of the PHQ-9, including a high coefficient alpha of 0.78, a high sensitivity of 85% and a specificity of 95%, render it suitable for global use [30]. The tool has been used across settings in sub-Saharan Africa and amongst different settings and racial groups [31–36], including in South Africa [37,38].

Ethical clearance to conduct the study was obtained from the SMU Research Ethics Committee (SMUREC/H/22/2021:PG) and permission was obtained from the Limpopo Department of Education. All participants provided informed consent. All COVID-19 safety regulations were adhered to at all times during data collection.

*3.4. Sampling*

Stratified sampling was conducted, whereby each of the schools in the Polokwane and Lepelle-Nkumpi local municipalities were divided into private and public schools, then into primary and high schools. Random sampling was conducted using the hat method to select names of schools from each category. The schools were then approached to request their participation, and all teachers who were willing to participate were recruited.

*3.5. Sample Size Determination*

Using the Raosoft sample size calculator for an estimated 10,000 teachers in the Capricorn District, a 5% margin of error, a confidence level of 95% and a response rate of 50%, a minimum sample size of 370 was calculated.

*3.6. Data Analysis*

The raw data were recorded in Microsoft Excel, cleaned, coded and exported to Stata-14 for analysis. The sociodemographic data were analyzed descriptively and expressed as means, medians, modes, proportions and percentages. The prevalence of depression was determined using the score obtained from the PHQ-9 scale. Scores below 4 were categorized as not depressed, and scores of 5 and above were categorized as depressed. Scores of 5 and above were further classified as mild (5–9), moderate (10–14), moderately severe (15–19) or severe (20 and above). Numeric data such as age, number of teachers, number of learners in the school and number of years in the teaching profession were converted to categories to reduce the number of numerical options.

Pearson's chi-square test of association was used to explore the associations between a range of sociodemographic variables and depression symptoms as measured by PHQ scores ($p \leq 0.05$). As needed, categorical variables were encoded using numerical codes in order to enable the performance of correlation with the categories of anxiety symptoms. A multivariate logistic regression model included the variables that were significantly associated with depression at a chi-square level of $p \leq 0.05$.

**4. Results**

A total of 25 schools participated in the study, including 11 (58.27%) primary schools, 13 (38.58%) high schools and 1 (3.15%) combined school. The majority of the participants (85.04%, $n = 324$) were employed in public schools, and 14.96% ($n = 57$) were employed in private schools. The majority (68.77%, $n = 262$) were located in Polokwane municipality, and 31.23% ($n = 119$) were located in Lepelle-Nkumpi municipality. The minimum participation rate in each school was 4, and the highest was 32. The average number of learners per school was 1039, covering schools from quintiles 1 to 5.

*4.1. Characteristics of the Teachers*

Table 1 reports the personal sociodemographic variables that were included in the analysis. The age of the participants ranged from 20 to 69, with a mean age of 41 years. The sample was predominantly Black (83.45%, $n = 318$); 70.87% ($n = 270$) or participants were female, and 29.13% ($n = 111$) were male. The greatest proportion (43.31%, $n = 165$) was either married or living with a partner, whereas 42.78% ($n = 163$) of participants we single, 7.09% ($n = 27$) were widowed and 6.82% ($n = 26$) were divorced. The majority (53.95%, $n = 205$) had a bachelor's degree as their highest qualification. Only 6.09% of participants had sought professional mental help in the previous 6 months.

*4.2. Employment-Related Factors*

Table 2 shows all the employment-related factors that were considered during analysis. The majority of the teachers were hired on a permanent basis (86.61%, $n = 330$), with just below half (46.19%, $n = 51$) of the teachers having been employed at their current school for over 6 years. The student-to-teacher ratio ranged from 15 to 50, with a mean of 36 students per class.

**Table 1.** Personal sociodemographic characteristics of educators.

| Variable | Frequency (n) | Percentage (%) |
|---|---|---|
| *Age (n = 381)* | | |
| Below 42 years | 193 | 50.66 |
| Above 42 years | 188 | 49.34 |
| *Gender (n = 381)* | | |
| Female | 270 | 70.87 |
| Male | 111 | 29.13 |
| *Race (n = 381)* | | |
| Black | 318 | 83.46 |
| Colored | 1 | 0.26 |
| Indian | 1 | 0.26 |
| White | 61 | 16.01 |
| *Home language (n = 381)* | | |
| Sepedi | 224 | 58.79 |
| Tsonga | 52 | 13.65 |
| Afrikaans | 51 | 13.39 |
| English | 14 | 3.67 |
| Zulu | 7 | 1.84 |
| Venda | 6 | 1.57 |
| Ndebele | 6 | 157 |
| Swati | 5 | 1.31 |
| Setswana | 4 | 1.05 |
| Xhosa | 1 | 0.26 |
| Other | 2 | 0.52 |
| *Marital status (n = 381)* | | |
| Married | 165 | 43.3 |
| Single | 163 | 42.78 |
| Divorced | 26 | 6.82 |
| Widowed | 27 | 7.09 |
| *Highest level of education (n = 381)* | | |
| Diploma | 113 | 29.74 |
| Bachelor's | 205 | 53.95 |
| Postgraduate diploma | 53 | 13.95 |
| Master's | 9 | 2.37 |
| *Consulted professional for mental health in past 6 months (n = 381)* | | |
| Yes | 23 | 6.04 |
| No | 358 | 93.96 |
| *Impact of COVID-19 on mental health (n = 381)* | | |
| Yes | 172 | 45.1 |
| No | 209 | 54.86 |

**Table 2.** Employment-related factors.

| Variable | Frequency (n) | Percentage (%) |
|---|---|---|
| *Teachers per school (n = 381)* | | |
| Below 33 | 213 | 55.91 |
| Above 33 | 168 | 44.09 |
| *Conditions of employment (n = 381)* | | |
| Permanent | 330 | 86.61 |
| Temporary | 51 | 13.39 |
| *Level of appointment (n = 371)* | | |
| Teacher | 313 | 86.46 |
| Head of department | 30 | 8.29 |
| Deputy principal | 14 | 3.87 |
| Principal | 5 | 1.38 |

**Table 2.** *Cont.*

| Variable | Frequency (n) | Percentage (%) |
|---|---|---|
| *Number of years as teacher (n = 381)* | | |
| 13 years and less | 191 | 50.13 |
| More than 13 years | 190 | 49.87 |
| *Years at current school (n = 381)* | | |
| 6 years and less | 205 | 53.81 |
| More than 6 years | 176 | 46.19 |
| *Number of subjects teaching (n = 381)* | | |
| 1–2 subjects | 216 | 56.70 |
| 3 or more subjects | 165 | 43.30 |
| *Learners per teacher (n = 381)* | | |
| Below 36 students per class | 179 | 46.98 |
| Above 36 students per class | 202 | 53.02 |
| *Subjects taught (n = 381)* | | |
| Languages | 177 | 46.46 |
| Mathematics | 136 | 35.70 |
| Commercial | 26 | 6.82 |
| Art | 31 | 8.14 |
| Life orientation | 89 | 23.36 |
| NS/tech | 35 | 9.19 |
| Social science | 39 | 10.24 |
| Other | 113 | 29.66 |
| *Quintile* | | |
| Q1 | 26 | 6.82 |
| Q2 | 75 | 19.69 |
| Q3 | 141 | 37.01 |
| Q4 | 8 | 2.10 |
| Q5 | 74 | 19.42 |
| Not applicable | 57 | 14.19 |
| *School* | | |
| Bokamoso | 16 | 4.20 |
| Capricorn | 10 | 2.62 |
| CM Sehlapelo | 8 | 2.10 |
| Dr Dixion Mphahlele | 31 | 8.14 |
| Dr MJ Madiba | 21 | 1.31 |
| Flora Park | 8 | 2.10 |
| Jabez Christian Academy | 12 | 3.15 |
| Kgwadu | 30 | 7.87 |
| Lebowakgomo | 9 | 2.36 |
| Makgongoana | 5 | 1.31 |
| Makgothane | 26 | 6.82 |
| Marara Cynthia | 29 | 7.61 |
| Masedibu | 12 | 3.15 |
| Matlalaohle | 32 | 8.40 |
| Mosepedi | 28 | 7.35 |
| Ngoatotlou | 4 | 1.05 |
| Northern Academy | 7 | 1.84 |
| PCS | 27 | 7.09 |
| PEMPS | 12 | 3.15 |
| Peter Nchabeleng | 6 | 1.57 |
| Phuti Makibelo | 4 | 1.05 |
| Piet Hugo Laerskool | 24 | 6.56 |
| Setototwane | 5 | 1.31 |
| Wonderland | 9 | 2.36 |
| Mogodumo | 21 | 5.51 |

### 4.3. Prevalence of Depression Symptoms

In response to the first research question that sought to determine the prevalence of depression symptoms among the sample, 190 (49.87%) teachers tested positive for symptoms of depression, with a majority categorized as mild (50.53%, $n = 96$) symptoms, followed by moderate (30%, $n = 57$), with only a few displaying moderately severe (12.10%, $n = 23$) and severe (7.37%, $n = 14$) symptoms, as shown in Figure 1.

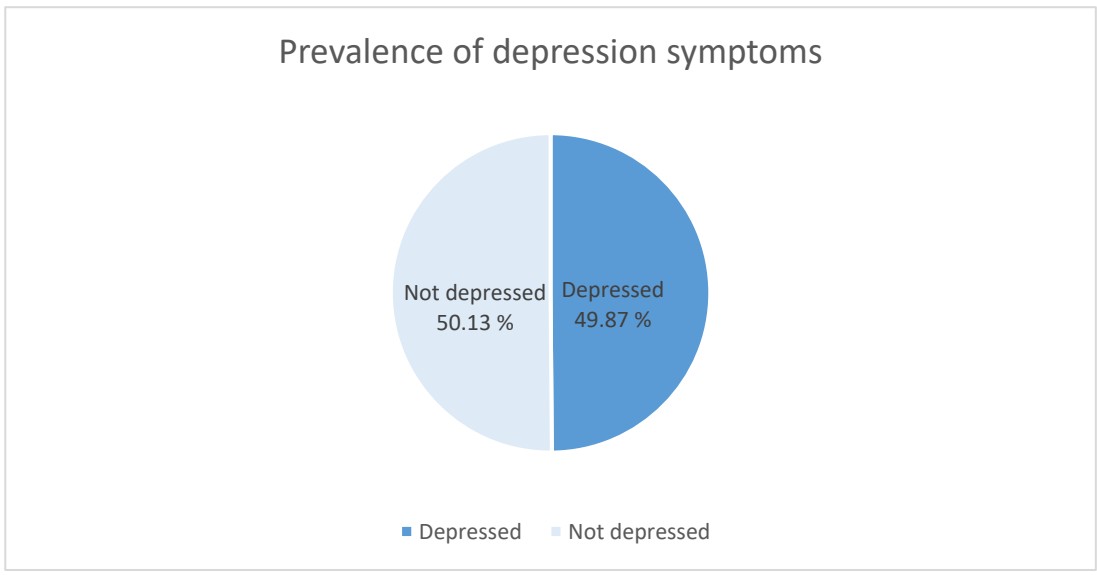

**Figure 1.** Pie chart showing the prevalence of depression.

*4.4. Reported Components of Depressive Symptoms*

A majority of the teachers in this sample reported symptoms of fatigue (60.30%, *n* = 230), disinterest and diminished pleasure in activities (52.49%, *n* = 200), sleeping problems (50.92%, *n* = 194), and feeling down and hopeless (190, *n* = 49.87%). Furthermore, 35.07% (*n* = 126) of the sample reported that they found it difficult to perform their day-to-day tasks as a result of the symptoms associated with depression, including tasks such as work and household duties, as well as managing interpersonal relationships. Some participants found it very difficult (6.56%), and fewer (0.52%) found it extremely difficult.

*4.5. Factors Associated with Depression Symptoms*

Table 3 shows the factors that were found to be significant in bivariate analysis. In response to the research question as to which personal sociodemographic factors were associated with depression symptoms, the Pearson chi-square test of association established that gender, marital status, race and whether the participant sought professional help with regard to mental health in the previous six months were significantly ($p \leq 0.05$) related to depression symptoms. The test further revealed a significant ($p \leq 0.05$) relationship between the impact of COVID-19 and symptoms of depression. For the question on which employment-related factors were associated with depression symptoms, the quintile ranking, subject taught, learner-to-teacher ratio and the school of employment were also significantly ($p \leq 0.05$) associated.

**Table 3.** Factors associated with depression.

| Factor | Frequency (%) | Depressed (%) | Not Depressed (%) | Chi$^2$ | *p*-Value |
|---|---|---|---|---|---|
| *Gender* | | | | 6.5558 | *0.010* |
| Female | 270 (70.87) | 146 (76.84) | 124 (64.92) | | |
| Male | 111 (29.13) | 44 (23.16) | 67 (35.08) | | |
| *Race* | | | | 9.4375 | *0.024* |
| Black | 318 (83.46) | 148 (77.89) | 170 (89.01) | | |
| White | 61 (16.01) | 40 (21.05) | 21 (10.99) | | |
| Colored | 1 (0.26) | 1 (0.26) | 0 (0.00) | | |
| Indian | 1 (0.26) | 1 (0.26) | 0 (0.00) | | |

**Table 3.** *Cont.*

| Factor | Frequency (%) | Depressed (%) | Not Depressed (%) | Chi² | *p*-Value |
|---|---|---|---|---|---|
| *Marital status* | | | | 10.4780 | *0.015* |
| Married | 165 (43.31) | 80 (42.11) | 85 (44.50) | | |
| Single | 163 (42.78) | 89 (46.84) | 74 (38.74) | | |
| Divorced | 26 (6.82) | 15 (7.89) | 11 (5.76) | | |
| Widowed | 27 (7.09) | 6 (3.16) | 21 (10.99) | | |
| *Consulted professional for mental health in past 6 months (n = 381)* | | | | 3.7985 | *0.051* |
| Yes | 23 (6.04) | 16 (8.42) | 7 (3.66) | | |
| No | 358 (93.96) | 174 (91.58) | 184 (96.34) | | |
| *Impact of COVID-19 on mental health (n = 381)* | | | | 36.2113 | *0.000* |
| Yes | 172 (45.1) | 115 (60.53) | 57 (29.84) | | |
| No | 209 (54.86) | 75 (39.47) | 134 (70.16) | | |
| *Learners per teacher (n = 381)* | | | | 5.3488 | *0.021* |
| Below 36 students per class | 179 (46.98) | 78 (41.05) | 101 (52.88) | | |
| Above 36 students per class | 202 (53.02) | 112 (58.95) | 90 (47.12) | | |
| *Subjects taught = social sciences* | | | | 10.4234 | *0.001* |
| Yes | 39 (10.24) | 29 (23.2) | 10 (5.24) | | |
| No | 342 (89.76) | 161 (84.74) | 181 (94.76) | | |
| *School quintile ranking* | | | | 11.5492 | 0.021 |
| Q1 | 26 (6.82) | 11 (11.58) | 15 (7.85) | | |
| Q2 | 75 (19.69) | 34 (17.89) | 41 (21.47) | | |
| Q3 | 141 (37.01) | 70 (36.84) | 71 (37.17) | | |
| Q4 | 8 (2.10) | 4 (2.11) | 4 (2.09) | | |
| Q5 | 74 (19.42) | 49 (25.79) | 25 (13.09) | | |
| Not applicable | 57 (14.19) | 22 (11.58) | 35 (61.40) | | |
| *School* | | | | 45.6868 | 0.005 |
| Bokamoso | 16 (4.20) | 11 (5.79) | 5 (2.62) | | |
| Capricorn | 10 (2.62) | 6 (3.16) | 4 (2.09) | | |
| CM Sehlapelo | 08 (2.10) | 4 (2.11) | 4 (2.09) | | |
| Dr Dixion Mphahlele | 31 (8.14) | 15 (7.89) | 16 (8.38) | | |
| Dr MJ Madiba | 21 (1.31) | 1 (0.53) | 4 (2.09) | | |
| Flora Park | 08 (2.10) | 4 (2.11) | 4 (2.09) | | |
| Jabez Christian Academy | 12 (3.15) | 4 (2.11) | 8 (4.19) | | |
| Kgwadu | 30 (7.87) | 16 (8.42) | 14 (7.33) | | |
| Lebowakgomo | 09 (2.36) | 6 (3.16) | 3 (1.57) | | |
| Makgongoana | 05 (1.31) | 3 (1.58) | 2 (1.05) | | |
| Makgothane | 26 (16.82) | 11 (5.79) | 15 (7.85) | | |
| Marara Cynthia | 29 (7.61) | 6 (3.16) | 23 (41.5) | | |
| Masedibu | 12 (7.61) | 5 (2.63) | 7 (3.66) | | |
| Matlalaohle | 32 (8.40) | 14 (7.37) | 18 (9.42) | | |
| Mosepedi | 28 (7.35) | 14 (7.37) | 14 (7.33) | | |
| Ngoatotlou | 04 (1.05) | 0 (0.00) | 4 (5.5) | | |
| Northern Academy | 07 (1.84) | 4 (2.11) | 3 (1.57) | | |
| PCS | 27 (7.09) | 17 (8.95) | 10 (5.24) | | |
| PEMPS | 12 (3.15) | 6 (3.16) | 6 (3.14) | | |
| Peter Nchabeleng | 06 (1.57) | 5 (2.63) | 1 (0.52) | | |
| Phuti Makibelo | 04 (1.05) | 2 (1.05) | 2 (1.05) | | |
| Piet Hugo Laerskool | 24 (6.56) | 20 (10.53) | 5 (2.62) | | |
| Setototwane | 05 (1.31) | 2 (1.05) | 3 (1.57) | | |
| Wonderland | 9 (2.36) | 8 (4.21) | 1 (0.52) | | |
| Mogodumo | 21 (5.51) | 6 (3.16) | 15 (7.33) | | |

*4.6. Final Multivariate Logistic Regression Model for Depression*

A logistic regression model was built using the nine factors identified by the association test. Only four factors remained significant (*p*-value ≤ 0.05) after multivariate analysis,

i.e., gender, marital status, subjects taught and impact of COVID-19. Further details are outlined in Table 4 below:

**Table 4.** Logistic regression model for depression.

| Factor | Coef. | Std. Err. | $P > |z|$ | 95% Conf. Interval | |
|---|---|---|---|---|---|
| Gender | −0.5545982 | 0.25329 | 0.029 | −1.051037 | −0.058159 |
| Race | 0.2436571 | 0.1342909 | 0.070 | −0.0195482 | 0.5068625 |
| Marital status | −0.313935 | 0.1371384 | 0.022 | −0.5827214 | −0.0451486 |
| COVID-19 impact | 1.324055 | 0.2324096 | 0.000 | 0.8685407 | 1.77957 |
| Sought professional mental health assistance in the past 6 months | 0.3348101 | 0.5162231 | 0.517 | −0.6769686 | 1.346589 |
| Learners per class | 0.3355893 | 0.2320937 | 0.148 | −0.1193059 | 0.7904845 |
| Quintile ranking | 0.0051952 | 0.0884746 | 0.953 | −0.1682119 | 0.1786024 |
| School of employment | 0.0101403 | 0.0185612 | 0.585 | −0.0262391 | 0.0465196 |
| Subject taught: social sciences | 1.329166 | 0.4167126 | 0.001 | 0.5124248 | 2.145908 |

## 5. Discussion

The gender representation of the sample is consistent with other studies that reported that the majority of teachers in the basic education sector are females [39,40]. The race of the participants is reflective of the reality of South African history, and the geographical location reflects the race of participants. The majority of the teachers held a bachelor's degree and had been in the teaching profession for 13 years or longer, which is consistent with other studies [5,7,41–43] that reported that most teachers tend to remain in the teaching profession for a long period. On the other hand, the longer teachers stay in the teaching profession, the more likely they are to experience occupational stress [7]. Occupational stress is a major risk factor for job dissatisfaction [44], as well as poor mental health outcomes such as depression [45].

The main aim of this study was to determine the prevalence of depression among teachers in Capricorn District, as well as the associated personal sociodemographic and employment-related factors. The study revealed a prevalence of 49.87% for depression symptoms among this sample of teachers, which suggests a high prevalence of previously undiagnosed depression symptoms, corresponding to almost half of the sample. This suggests that mental health disorders, including depression, are still largely undiagnosed and thus not treated [46,47]. The high prevalence is also similar to findings reported in other countries, such as Tanzania, Malaysia and Chile, were depression symptoms were reported among 51%, 67.3% and 43.3% of teachers, respectively [6,7,48], which seems to support the notion that the teaching environment in various countries is associated with various stressors that predispose teachers to mental disorders, including depression. This high prevalence is associated with high economic costs, which include increased use of medical care, lower quality of life and decreased workplace productivity [49].

The finding that only 6.09% of the sample had sought professional mental health assistance in the previous 6 months reflects poor help-seeking behavior, even among those who know that something is wrong within them. Poor acknowledgment and help-seeking behavior for mental illness has been previously reported and can be explained by a lack of understanding of the symptoms of mental illness [50] and denial of symptoms due to the stigma attached to mental illness [51]. Such stigma makes a difference between those who seek treatment and those who do not.

One of the research questions was aimed at identifying the personal sociodemographic factors that were associated with depression symptoms. Personal sociodemographic factors such as gender and marital status were significantly associated with depression, and female teachers and those who were single presented heightened depression symptoms, which

was previously reported in other studies [8,48,52,53]. Female teachers are more likely to be emotionally responsive and take on caregiving roles with their leaners as compared to male teachers [54], which not only leads to increased responsibilities outside the scope of their work but greatly contributes to teacher role ambiguity and leads to elevated stress levels, making them more susceptible to depression.

The current study revealed that single teachers were at an increased risk of depression as compared to teachers who were married. The literature reports that marriage can be a protective factor due to its ability to provide companionship and stress alleviation [55], which aids in social support [18]. Scholars have illustrated this by suggesting benefits such as shared parental responsibilities, financial stability and emotional support [56] as some of the reasons why marriage has been associated with positive mental health outcomes [53–56]. Moreover, loneliness has been indicated as a risk factor for depression, and single teachers may have felt this burden even more as a result of the COVID-19 pandemic [57]. For example, this study illustrates that 45% of depression symptoms were attributable to the COVID-19 pandemic. It has been reported that the challenges brought forth by the pandemic exacerbated the burden of depression globally [58], including in the school environment, in which the pandemic introduced abrupt changes [4,58–60]. The shared global concerns regarding fear of infection, job insecurity, loss of income and loss of life also affected educators throughout the various waves of the pandemic. Teachers had to juggle their work responsibilities in the midst of uncertainty, fear and risk of infection, which elevated psychological distress [61]. The added responsibilities of online teaching during the sudden closures of school took a toll on educators and students alike, especially those teaching in economically disadvantaged schools, as problems such as a lack of resources, issues with connectivity and inadequate technological knowledge presented many problems with respect to effective online learning during this time [62–64]. Additionally, after the reopening of schools, there were extra duties that teachers had to perform to ensure safety—for example, checking temperatures, teaching a smaller amount of children per class and other social distancing practices—on top of their normal duties, which contributed to increased stress levels and therefore depression [59–61].

With regard to the research question that was aimed at identifying the employment factors associated with depression, the current study identified several employment-related factors, including quintile ranking, which is the system that divides South Africa's public schools into five quintile rankings. The indices for such rankings are based on the income, literacy and unemployment levels in a community, which determine the socioeconomic status of the said community. Schools in quintiles one to three are no-fee-paying, whereas schools in quintiles four and five are ranked as fee-paying schools. The majority of schools ranked one to three are situated in rural areas or previously marginalized communities, and quintiles four and five are located in affluent neighborhoods [14]. The current study revealed that compared to teachers in higher-quintile-ranking and private schools, teachers in lower-quintile-ranking schools (Q1–Q3) were more likely to display symptoms of depression. This confirms that low socioeconomic status of both the individual and the community are risk factors for depression [65–67].

The current study also identified the learner-to-teacher ratio as significantly associated with the development of depression symptoms; it has been previously reported that larger classes of learners put extra demands on teachers, increasing the risk for depression [68]. In South Africa, affluent communities contribute to favorable learner–teacher ratios by paying for extra teachers over and above those provided by the government. Although South African legislative guidelines indicate that the learner–teacher ratio is supposed to be 33 to 1 [69,70], this study revealed ratios of up to 50 leaners per teacher in some schools. This highlights the extra workload and pressure placed on teachers and further implies that learners are not receiving the individual attention that is required from their teacher [70]. The aforementioned factors are usually common in schools situated in poverty-stricken areas, as issues surrounding overcrowding, lack of resources and lack of infrastructure are more rampant [63,70,71]. It has been illustrated in previous studies that such issues

predispose teachers to negative mental health outcomes, which explains the high levels of depression [2,23,68]. Thus, factors within individual schools play a contributory role in poor mental health outcomes, which explains why depression may be more prevalent in one particular school as opposed to another.

While other studies have found no association between subjects taught and ill mental health [72,73], this study revealed that compared to other teachers, teachers who taught social sciences were at a lesser risk for depression symptoms. Scholars have argued that teachers of core subjects such as mathematics and languages have a higher volume of students, which leads to an increased workload and therefore heightened stress levels [74]. However, it is worth noting that the area of subjects taught and depression remain limited; therefore, further research is required.

The majority of the sample presented mild (25.20%) to moderate (14.96%) symptoms, and a majority (64.83%) of the teachers indicated that symptoms interfered with their daily activities. The most commonly reported problems were fatigue (60.30%); disinterest in daily activities (52.49%); hopelessness (49.87%); and eating (50.92%), sleeping (50.92%) and concentration (38.85%) problems. This result reinforces what was previously reported in other studies that identified sleeping difficulties [75], eating problems [26] and fatigue [76] as common problems experienced amongst educators. Even more concerning is the fact that 11.55% of educators reported having thoughts of self-harm. Depending on the intensity of these problems, they often interfere with interpersonal relationships and make it difficult to engage in work or academic activities. Previous studies reported lower levels of productivity and high levels of absenteeism in association with depression amongst educators, which lead to decreased quality of learning [2]. Some cross-sectional studies have even indicated depression amongst educators as a potential risk factor for depression amongst students under their care [27]. Others have found reduced academic, social and emotional development amongst students being taught by teachers who are depressed [77–79]. Glazzard and Rose [80] further reported that students could pick up whether their teachers were stressed, irrespective of how well teachers tried to hide it. It is therefore necessary to address depression within the school setting, not solely for the benefit of teachers but for the well-being of leaners as well.

## 6. Conclusions and Limitations

There are various work-related stressors and personal factors that contribute to the high prevalence of depression amongst this sample of teachers. However, the consequences of not addressing the mental health status of educators does not only affect them but their students and overall quality of education, making mental health intervention among teachers of empirical importance. The literature has reported that depression among teachers has negative impacts on the academic development of their students, which implies that the impact can be long-lasting. A lack of attention to teachers' mental health is therefore a risk for the next generation of the learners and needs to be addressed as a matter of urgency. It is recommended that the wellness programs of the Department of Basic Education be intentional in integrating mental health components that will be custom-made to respond to the mental health of teachers. Such interventions should be ongoing as part of mental health promotion of teachers.

The inclusion of many categories under "school name" may have affected the results of this study, which is regarded as a limitation. Additionally, the study was conducted during the COVID-19 pandemic, which could have impacted on the results, as the pandemic has been reported to impact the mental health of people. The data collection method had to be altered in order to adhere to the COVID-19 safety and regulation protocols, which meant that the researcher relied on the interest of school management teams for staff participation, as she could not personally address the teachers to recruit them for the study, which may have affected the response rate.

**Author Contributions:** Conceptualization, K.E.M.; formal analysis, K.C.M., investigation, K.C.M.; data curation, K.C.M., writing—original draft preparation, K.C.M.; writing—review and editing, K.E.M., supervision, K.E.M. All authors have read and agreed to the published version of the manuscript.

**Funding:** This study was jointly funded by the National Research Foundation (115449) through the Research Chair: Substance Abuse and Population Mental Health grant and the South African Medical Research Council (M052) through the adolescent mental health grant.

**Institutional Review Board Statement:** Ethical clearance to conduct the study was obtained from the SMU Research Ethics Committee (SMUREC/H/22/2021: PG) and permission was obtained from the Limpopo Department of Education.

**Informed Consent Statement:** All participants provided written informed consent.

**Data Availability Statement:** Data is contained within the supplementary material provided to MDPI. If requested, the data presented in this study can be available following the data availability policies of Sefako Makgatho Health Sciences University.

**Acknowledgments:** Gratitude is extended the Limpopo Department of Education for providing permission for data collection, as well as the principals, deputy principals, management teams and individual teachers who participated in the study.

**Conflicts of Interest:** The authors declare no conflict of interest.

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
