# Peer review of "Is the Teaching Environment a Risk Factor for Depression Symptoms? The Case of Capricorn District in Limpopo, South Africa"

_education, doi:10.3390/educsci13060598_

Round 1

Reviewer 1 Report

This manuscript presents the findings of an intriguing study on the level of depression in teachers working in schools in the Capricorn District of South Africa's Limpopo Province.

The manuscript is well-organized, but it could be much better. The research questions are unclear and unjustified. The literature review and research design ignore a large body of research that used the Maslach Burnout Inventory (MBI). My recommendation is to improve the literature review, and if the authors agree, they can defend their decision to use the Patient Health Questionnaire by referring to the work of Williamson, et al 2018. https://doi.org/10.4300%2FJGME-D-18-00155.1 who found that MBI performed similarly to other instruments like the Patients Health Questionnaire.

Another issue I have with this manscript is that the research design ignored several relevant external and internal school parameters, such as family income and other demographics of the parents.
The text's flow must be improved, beginning with the general and gradually focusing on Africa in terms of the profession's historical problems, the level of professional burnout, and the parameters that can lead to depression.

The work's objectives, as presented in the introduction, are at odds with the data and discussion. I am also concerned that the authors failed to justify the sampling methodology and research tools used in this work.
For example, one significant parameter for workload was school size, and school conflict was ignored during the sampling. Similarly, the authors used the Patient Health Questionnaire, but their research was limited and did not include additional instruments or multivariate analyses. The authors' work could be improved by following the research and analysis method used in a recently published similar work https://doi.org/10.3389/fpsyt.2021.644276.
In any case, the statistical analysis must be consistent with the research questions, and the significance of differences between demographic groups must be clearly demonstrated in the results.

Reviewer 2 Report

Thank you for the opportunity to review this article. The article is well written, the text is clear and easy to read. The topic ‘determining the prevalence of depression amongst teachers, as well as contributing factors, in the Limpopo Province’, is relevant and important in the context of education and mental health. The question posed by the researchers is important and interesting. The argument is clear, followed through and supports the conclusion. The cross-sectional design using self-administrated questionnaires are appropriate, and the data supports the conclusion. Ethical aspects were considered, adhered to, and arrangements were made during the Covid-19 pandemic to deal with challenges such as access to the schools. The findings are of value both for the South African, African and international context.

Minor technical aspects:

Line 121:  A researcher developed the socio-demographic that questionnaire was used to collect demographic data from participants.  The sentence is unclear.

Line 145: Please insert a full stop after the sentence …for a score of 20 and above. Numerical data

Author Response

Good day, 

Kind regards
